# Imputation of 3D genome structure by genetic–epigenetic interaction modeling in mice

**Lauren Kuffler, Daniel A Skelly, Anne Czechanski, Haley J Fortin, Steven C Munger, Christopher L Baker, Laura G Reinholdt, Gregory W Carter\***

The Jackson Laboratory, Bar Harbor, United States

**\*For correspondence:**
Gregory.Carter@jax.org

**Competing interest:** The authors declare that no competing interests exist.

**Abstract** Gene expression is known to be affected by interactions between local genetic variation and DNA accessibility, with the latter organized into three-dimensional chromatin structures. Analyses of these interactions have previously been limited, obscuring their regulatory context, and the extent to which they occur throughout the genome. Here, we undertake a genome-scale analysis of these interactions in a genetically diverse population to systematically identify global genetic–epigenetic interaction, and reveal constraints imposed by chromatin structure. We establish the extent and structure of genotype-by-epigenotype interaction using embryonic stem cells derived from Diversity Outbred mice. This mouse population segregates millions of variants from eight inbred founders, enabling precision genetic mapping with extensive genotypic and phenotypic diversity. With 176 samples profiled for genotype, gene expression, and open chromatin, we used regression modeling to infer genetic–epigenetic interactions on a genome-wide scale. Our results demonstrate that statistical interactions between genetic variants and chromatin accessibility are common throughout the genome. We found that these interactions occur within the local area of the affected gene, and that this locality corresponds to topologically associated domains (TADs). The likelihood of interaction was most strongly defined by the three-dimensional (3D) domain structure rather than linear DNA sequence. We show that stable 3D genome structure is an effective tool to guide searches for regulatory elements and, conversely, that regulatory elements in genetically diverse populations provide a means to infer 3D genome structure. We confirmed this finding with CTCF ChIP-seq that revealed strain-specific binding in the inbred founder mice. In stem cells, open chromatin participating in the most significant regression models demonstrated an enrichment for developmental genes and the TAD-forming CTCF-binding complex, providing an opportunity for statistical inference of shifting TAD boundaries operating during early development. These findings provide evidence that genetic and epigenetic factors operate within the context of 3D chromatin structure.

## eLife assessment

This **important** manuscript reports interactions between genetic variation, DNA accessibility, and chromatin structure in gene expression at a genome wide scale. The authors found that most of these interactions occur within topologically associating domains (TADs) and 3D genome structure data can be efficiently used to guide the discovery of significant genetic and epigenetic influences on gene expression. Overall, this **convincing** study highlights the importance of 3D chromatin structure in controlling how gene expression is regulated by genetic and epigenetic processes.

## Introduction

Local regulatory mechanisms within the genome and their interaction with chromatin structure give rise to subtle variations in gene expression. Yet the interacting effects that genetic and epigenetic factors produce on gene transcription are rarely studied at a genome-wide scale, leaving us without global information on a key step between the genetic code and the phenotype. Studies are generally limited to examination of individual regions or overlapping single-nucleotide polymorphisms (SNPs) and open chromatin peaks with limited investigation into how these regulatory elements combine to affect gene transcription (*Ackermann et al., 2016*; *Wu et al., 2016*). Genetic analyses are a powerful approach that allows the study of these interactions. Conversely, phenotypic variation in genetically diverse populations is a result of both genetic and epigenetic factors operating in tandem. Understanding the scope and landscape of these interactions on a genome-wide scale is a vital step toward deciphering the genetic regulation of gene expression and, in turn, the mechanisms of non-coding variation on phenotypic outcomes.

Physical distance along a linear genome is a common metric for determining whether a putative regulatory element will affect a given gene's transcription. This has led to the concept of a 'local area' in which most regulatory interactions take place. Many have used a 100- to 500-kb flanking window around a gene to encapsulate the landscape of local epigenetic effects (*Veyrieras et al., 2008*; *Stranger et al., 2007*; *Dixon et al., 2007*). This interval size is motivated more by convenience than rigorous evidence. Incidentally, the estimated scale of this local area is approximately equivalent to the size of a topologically associated domain (TAD).

TADs are chromatin loops with an average length of approximately 1.1 Mb that are joined via the architectural protein CTCF and the cohesin protein complex at known binding motifs (*Dixon et al., 2012*; *Pombo and Dillon, 2015*). TADs are believed to be mostly cell-type invariant and conserved across species (*Dixon et al., 2012*; *Harmston et al., 2017*; *Yakushiji-Kaminatsui et al., 2018*). The current hypothesis is that TADs provide physical boundaries for local regulatory function, but functional analysis of regulatory interactions within TADs has been limited by the lack of informative genetic variation in the studies that led to TAD discovery. Reverse genetic approaches like saturation mutagenesis can be used to introduce functionally informative genetic variation. But such screens are difficult in mammals, and even modern CRISPR–Cas9-based approaches are still limited in their scope and application outside of immortalized cell lines (*Serebrenik and Shalem, 2018*).

Samples from genetically diverse populations permit comprehensive study of complex genetic interaction and non-coding regulation on a genome-wide scale (*Svenson et al., 2012*). The Diversity Outbred (DO) mouse population is a heterogeneous stock composed of mice derived from eight founder strains, including three wild-derived strains (*Churchill et al., 2012*). The ongoing process of random outbreeding produces a mouse population segregating millions of precisely mapped SNPs, with well-balanced allele frequencies, and increasingly small blocks of linkage disequilibrium that now approach the size of TADs (*Broman, 2012*; *Chesler et al., 2016*). Samples from the DO have been used to create extensive resources and datasets, including mouse embryonic stem cell (mESC) lines. A recent study used mESCs derived from generation 19 DO embryos, grown in the absence of ERK inhibition to study the gene regulatory networks that stabilize pluripotent states in vitro. This revealed phenotypic variability in features of ground state pluripotency (*Skelly et al., 2020*).

Here, we determine the extent to which genetic–epigenetic interactions between single-nucleotide polymorphisms (SNPs) and regions of open chromatin are present, and uncover the biological basis of their distribution in three-dimensional (3D) genomic organization (*Figure 1*). Our results show that genetic–epigenetic interactions were found across the genome and involve regulatory elements that would not be identified with single-omics data. These interacting elements cluster together within TADs bounded by previously identified active CTCF-binding sites. We infer chromatin structure from interaction data, analyze interaction contributions from the main and interaction regression elements, identify potential regulatory functions underlying a set of interactions, and correlate these interaction behaviors to CTCF-binding differences in inbred founder lines.

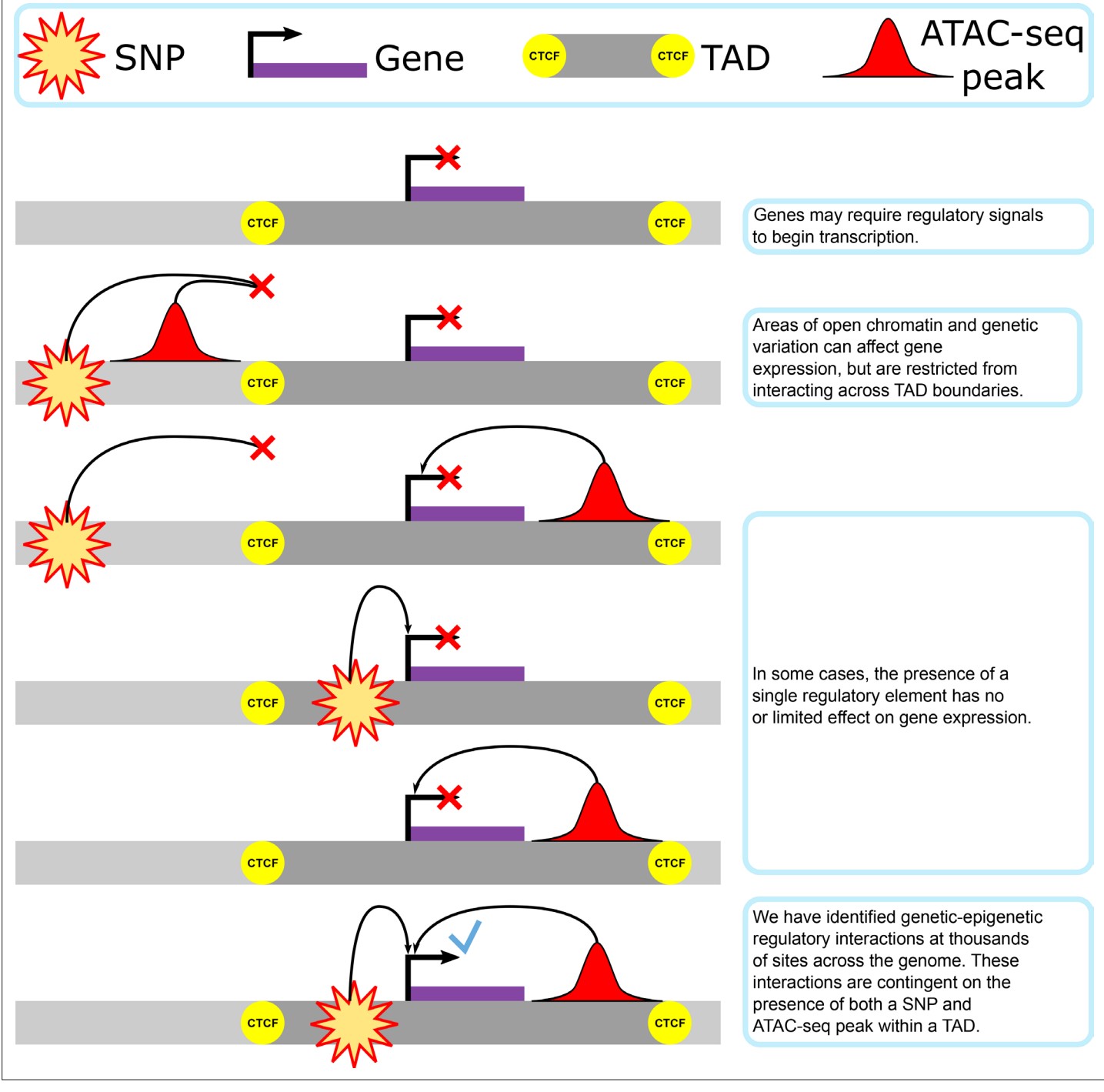

**Figure 1.** Our proposed model of genetic–epigenetic regulatory interaction, as bounded by topologically associating domains.

## Results

### Genetic–epigenetic interactions are pervasive and enriched within TAD boundaries

Although there are numerous reports detailing examples of genetic variants and chromatin state altering gene expression, the non-additive interactions between these features have yet to be systematically explored. We aimed to determine the extent to which genetic–epigenetic regulatory interactions occur on a systemic level, and whether these interactions exhibit any bias in their location within

the genome. To explore these interactions, we utilized paired transcriptome and chromatin accessibility profiling of DO mESCs (*Skelly et al., 2020*). These samples incorporate genetic backgrounds from eight well-characterized founder strains from three *M. musculus* subspecies, randomly outbred for 19 generations. Thus, the resultant data constitute a unique resource for addressing questions of genetic–epigenetic regulatory interactions.

To search for non-additive interaction effects on a global scale, we developed a method of analysis that could be broadly applied to, and could permit discrimination between, co-occurrence and interaction effects. To that end, we combined regression model fitting (see Methods, *Equation 1*) and model selection via information criteria in order to test whether gene expression was affected by a genetic variant, an open chromatin region, both independently of each other, or both working in concert.

To account for the resolution limits on genetic variant effects imposed by linkage disequilibrium, we subsetted our genotype data to 68,413 high confidence haplotype calls evenly spaced by genetic distance to reflect this resolution limit (Methods). We collated these with 102,173 ATAC-seq peaks and 13,631 genes with RNA-seq coverage within our sample set.

With 95 trillion possible interactions genome-wide, testing had to be focused to an informative and computationally feasible subset. To assess the significance and extent of non-additive interaction models and identify target locations for more focused analyses, a subset of randomly selected trios of intrachromosomal genes, genetic variants, and ATAC-seq peaks were tested with the interaction model. A total of 102,104 ATAC-seq peaks, 68,413 variants, and 13,631 genes were used to model 39,021,625 combinations. This resulted in 27,509 significant models (Bonferroni adjusted $p < 1 \times 10^{-7}$), or 0.07% of all models, involving 19,145 ATAC-seq peaks, 14,476 variants, and 1286 genes (18.75%, 21.16%, and 8.52% of the total input, respectively). Complete results are tallied in *Supplementary files 1 and 2*.

If interactions were detected randomly, we would expect a null distribution with no enrichment of low p-value models. Instead, the number of significant models was four orders of magnitude greater than expected, thus indicating the presence of true associations (*Figure 2A*). The locations of significant associations indicated non-additive interactions were much more likely to occur within the linkage disequilibrium block that contains the gene (*Figure 2B*), matching prior work that the majority of detectable genetic effects originate in local regulatory regions (*Ronald et al., 2005*; *Su et al., 2010*; *Nelson et al., 2004*). Approximately half (49%) of all significant interacting models included a genetic variant or ATAC peak within 4 Mb of the gene they affected. Within this range, we found that less than 10% of variants and ATAC peaks clustered closer to each other than they did to the gene they affected, indicating that co-localization may not the driving factor in many interacting models (*Supplementary file 3*).

As a result of these findings, we considered potential chromatin features that might influence the local area for genetic–epigenetic interactions. We hypothesized that TADs may play a role in determining the prevalence of non-additive interactions, as they are a 3D genome structure known to enable enhancer access to local genes. Therefore, we generated a second, TAD-focused subset of regression models, testing all potential combinations between each gene and every genetic variant and ATAC peak that fall within one TAD of its transcription start site (TSS) in either direction. TAD boundary locations were retrieved from publicly deposited Hi-C-based analyses of B6 embryonic stem cells (*Dixon et al., 2012*).

Models with at least one interacting element within the same TAD as the gene were 3.7 times more likely to reach our threshold, compared to models that did not (3.3% vs. 0.9%). This constituted 3836 genetic variants and 3725 ATAC-seq peaks affecting the expression of 896 genes in our initial exploratory subset, and 11,904 SNPs, 25,961 ATAC-seq peaks, and 1099 genes in our equivalently sized TAD-centric subset (*Supplementary files 1 and 2*).

ATAC-seq peaks involved in non-additive interactions favored results between CTCF-binding sites associated with TAD formation (*Figure 2—figure supplement 1A*). Regression models that did not identify interaction involvement also show this local, TAD-bound organization of ATAC-seq peaks (*Figure 2—figure supplement 1B*). Interacting regulatory elements as defined by our model are thus clustered within the same TAD as the gene they affect. This provides a possible structural explanation for previously observed local regulatory ranges.

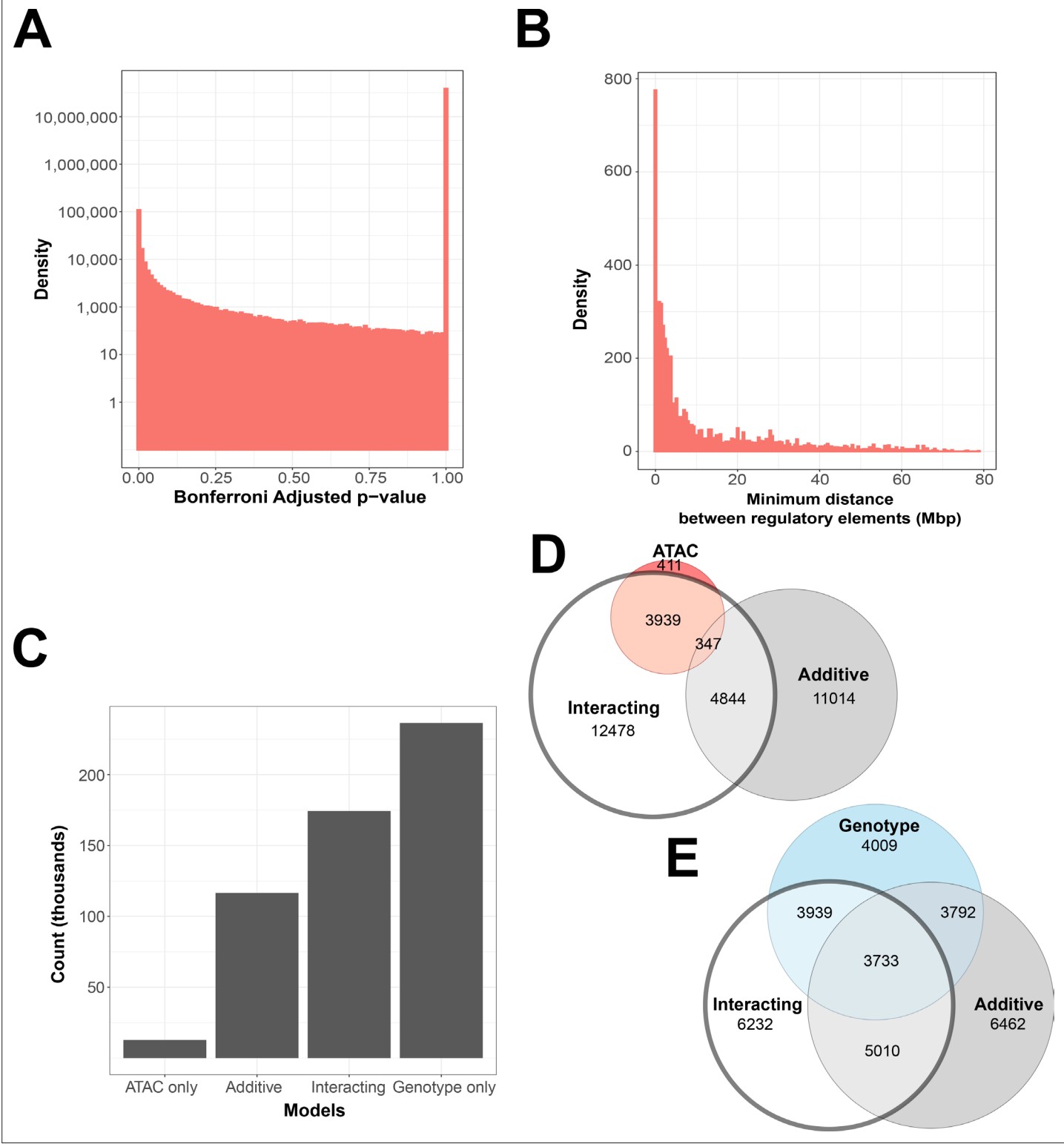

**Figure 2.** Interacting and additive models are abundant and favor local genes. (**A**) Density of Bonferroni adjusted p-value distribution in randomly sampled intrachromosomal models. (**B**) Density of non-additive interacting models where adj. $p < 1 \times 10^{-7}$, by minimum distance between regulatory elements. (**C**) Distribution of model term retention across intra-topologically associated domain (TAD) models. (**D, E**) Euler plots of ATAC-seq peaks and genetic variants, dividing them by their participation in single-term, additive, and non-additive interacting models. ATAC-seq peaks and genetic variants involved in interacting models are highlighted, as they are investigated further in this paper.

The online version of this article includes the following figure supplement(s) for figure 2:

*Figure 2 continued on next page*

*Figure 2 continued*

**Figure supplement 1.** Additional analysis of interacting and additive models and their relation to local genes.

## Interacting elements in genetically diverse samples escape conventional discovery methods

We next evaluated whether genetic–epigenetic, non-additive interactions provide information that genotyping or ATAC-seq alone cannot. To this end, we compared interacting models, found to represent 32.29% of significant intra-TAD models (*Figure 2C*), with models associated with genetic variants or ATAC-seq peaks without interactions. We broke down the overlap between intra-TAD regression model results by genetic variants and ATAC-seq peaks separately.

Previous studies have found correlation between local open chromatin and gene expression, including within this DO mESC population (*Skelly et al., 2020*). Mediation analysis has indicated more complex co-regulatory relationships are also extant, but their relationships have not been comprehensively studied.

Each genetic or epigenetic regulatory feature input to our regression model can potentially influence multiple genes, alongside multiple other regulators. Thus, each feature could be engaged in many different regulatory relationships with its local genes, some of which may or may not be detectable with correlation or mediation analyses. We wished to determine how many regulatory elements were exclusively found in interacting regression models, and would thus escape notice with conventional analyses.

We found that 26.35% of genetic variants are shared between non-additive interacting models and additive models, and 23.12% are shared with models that contain no ATAC-seq peak contribution. In contrast, few models contain *only* an ATAC-seq peak contribution, and 41.81% of ATAC-seq peaks are exclusive to interacting models (*Figure 2D*). This is also in contrast to results from our randomly sampled gene–ATAC–genotype trios, which show a preference for genotype-only models (*Figure 2—figure supplement 1C, D*).

These results suggest that although genetic variants are the primary driver of variation in expression, non-additive interactions with chromatin states further reveal the origins of expression variation. Our findings also suggest that genetic variation and open chromatin data cannot be used independently to capture these regulatory features. Conversely, this suggests that ATAC-seq data alone are a less effective predictor of gene expression in a genetically diverse population.

## TAD boundaries limit genetic–epigenetic interactions

The specific preference for non-additive interactions within active CTCF-binding sites warranted further study. We sought to determine if active CTCF-binding sites provided an appreciable boundary to interactions (*Figure 3A*), similar to reported segregation of enhancer elements (*Rodríguez-Carballo et al., 2020*). We carried out regression analyses across the genome for all possible models involving each gene–genotype–ATAC combination within the gene's TAD and nearest flanking TADs or intra-TAD regions. Although resolution of causal variant location limited by linkage disequilibrium, ATAC-seq peaks that interact with variants could be confidently localized in relation to TAD boundaries.

We found that TADs contained more ATAC peaks that interact with local genetic variants to affect expression of a resident gene (*Figure 3B*). With a window of 200 kb on either side of an active CTCF boundary, we found 52,511 ATAC-seq peaks sharing a TAD with their interacting gene, versus 24,229 peaks in different TADs or inter-TAD regions This constitutes a significant enrichment of intra-TAD gene–peak interactions over expectations based on overall ATAC-seq peak density (an odds ratio of 1.48, null expectation $p < 2.2 \times 10^{-16}$) (*Figure 3—figure supplement 1*). We also identified 29 individual ATAC-seq peaks that were involved in more than 200 non-additive interactions each, which are distinctly visible within the aggregate view (*Figure 3—figure supplement 2*). These were exclusively affecting genes within the same TAD as the interacting ATAC-seq peak, which suggests that our results are not due to higher density of open chromatin within TAD structures. This is consistent with previous findings that the effects of enhancers and other regulatory elements are constrained by TADs (*Krijger and de Laat, 2016*).

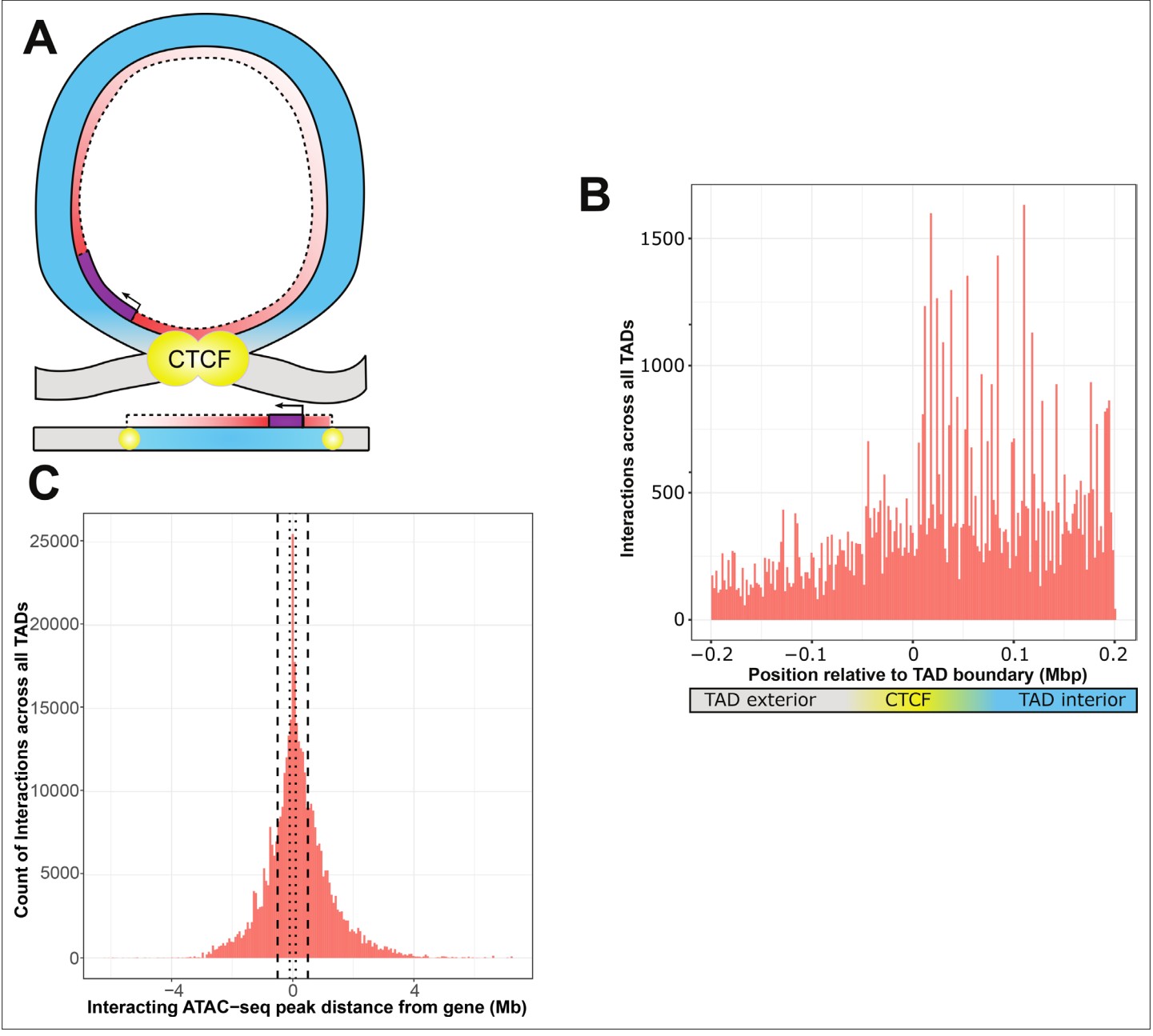

**Figure 3.** ATAC-seq peaks that interact with genetic variants generally reside within the affected gene's topologically associated domain (TAD). (**A**) Schematic of a TAD loop, including gene (purple) and density of interacting model elements (red). Loop interior is in blue, exterior DNA is gray, and CTCF-binding sites are in yellow. (**B**) Location of interacting ATAC-seq peaks relative to TAD boundary location, merged across all genes. TAD interior denotes the TAD in which the dependent gene was found. (**C**) Interacting ATAC-seq peaks by distance from associated gene transcription start site (TSS). Local area cutoffs of 100 and 500 kb flanking regions are marked.

The online version of this article includes the following figure supplement(s) for figure 3:

**Figure supplement 1.** Overall ATAC-seq peak distribution does not fully explain the distribution of interacting ATAC-seq peaks.

**Figure supplement 2.** Highly interacting ATAC-seq peaks are contained within the same topologically associated domain (TAD) as the genes they affect.

## Revising constraints on local regulator area using genetic data

Definitions of local regulatory area are used to establish the scope of analyses.

Many cell types and animal models do not have publicly available Hi-C or CTCF ChIA-PET datasets, further limiting many researchers that may want to study genetic–epigenetic non-additive interactions.

Our data support that TADs may act as a biologically defined boundary. Therefore, we wanted to determine how far the TAD-defined local area was likely to extend from any given gene. We found that to capture 95% of inter-TAD genetic–epigenetic interactions at adjusted $p < 1 \times 10^{-7}$, a window of 2,074,778 bp upstream and 2,575,581 bp downstream of the gene TSS was required (*Figure 3C*). The density of results using linear DNA sequence does not necessarily experience a linear drop-off over this distance, due to variable TAD lengths (*Figure 4E, F*).

We next considered the location of these interacting elements with regard to genes and known regulatory elements. ATAC-seq peaks involved in non-additive genetic–epigenetic interactions were analyzed by overlap with mm10 NCBI RefSeq genes and selected Enhancer Atlas 2 datasets, as well as FANTOM5's list of mm10 promoters (*Lizio et al., 2015*; *Supplementary files 4 and 5*). 65.37% of peaks on each chromosome fell within gene bodies. Normalized for length, exons were especially enriched, with 34.41% of peaks falling within exons, half of which fell within the first (or only) exon. 24.06% of peaks overlap with promoters. 6.09% of peaks fall within enhancers curated for ESC R, R1, KH2 embryonic stem cell lines. There was no inverse relationship between gene body versus enhancer peak percentage (*Figure 4—figure supplement 1A*), indicating that this was not simply reflecting the relative density of genes within various TADs across the genome.

## Density of interactions between genomic elements is defined by 3D context

If TADs act as a constraint to local interaction, then their 3D looping structure should be reflected in the regulatory patterns of genes found within them. At the most basic level of TAD structural organization, the CTCF-binding site brings distant areas of chromatin into close physical association. We analyzed genome-wide distribution of genotype-interacting ATAC peaks relative to genes within their respective TADs.

This revealed that the gene-centric increase in regulatory interaction density responded to the 3D context of the TAD loop, crossing CTCF boundaries irrespective of linear DNA sequence. This creates a distribution of non-additive interactions centered at the gene promoter that reaches its minimum halfway around the TAD loop from the gene (*Figure 4A*). In fact, we found local regulatory regions that displayed organization that spanned the TAD boundary sites, creating interaction density gradients that appear non-linear when viewed without TAD data (*Figure 4B*).

We sought to determine whether gene-centered, TAD-internal searches for interacting elements were potentially a more efficient method for searching for non-additive interactions, when compared to local area searches along the linear DNA sequence. We compared the two methods, running a pair of genome-wide, gene-centric analyses out to the limit of our TAD-internal search (±~2.5 Mb). We calculated the percentage of ATAC-seq peaks within this region that participated in significant interactions (*Figure 4C, D*). The overall density of ATAC-seq peaks within these regions was also compared (*Figure 4E, F*).

We found that TAD-based searches produce consistently higher percentages of interacting ATAC-seq peaks across the analysis, and higher ATAC-seq peak density out to ±~1 Mb. TAD-based searches experience a progressive falloff in ATAC-seq peak density when compared to linear sequence-based searches. This is largely due to dropout of smaller TADs from the analysis. This results in more variable interaction percentages past ~1 Mb from the gene.

Linear search results show a consistently lower percentage of interacting ATAC-seq peaks, and a lower density of ATAC-seq peaks per locus. This was unexpected, as the immediate vicinity of the gene was expected to contain a similar density and non-additive interaction potential of ATAC-seq peaks when compared to the TAD-based search method. We therefore hypothesized that TAD boundaries in close proximity to some genes might create this discrepancy. We found that while the majority of genes were randomly distributed within their TADs, 12,437 or 24,2124.31% of genes were found to have a TAD boundary close upstream to their TSS (*Figure 4—figure supplement 1B*).

Interestingly, there is a depletion in the non-additive interaction rate of ATAC-seq peaks flanking the gene (*Figure 4D*), which is not noted in raw interaction counts (*Figure 4A*). This may be due to the necessary presence of open chromatin at the promoter and the gene body during priming and transcription regardless of interaction. There are also genes which are concomitantly active within our samples, and do not show variation in expression, thus experiencing activity without detectable regulatory interaction.

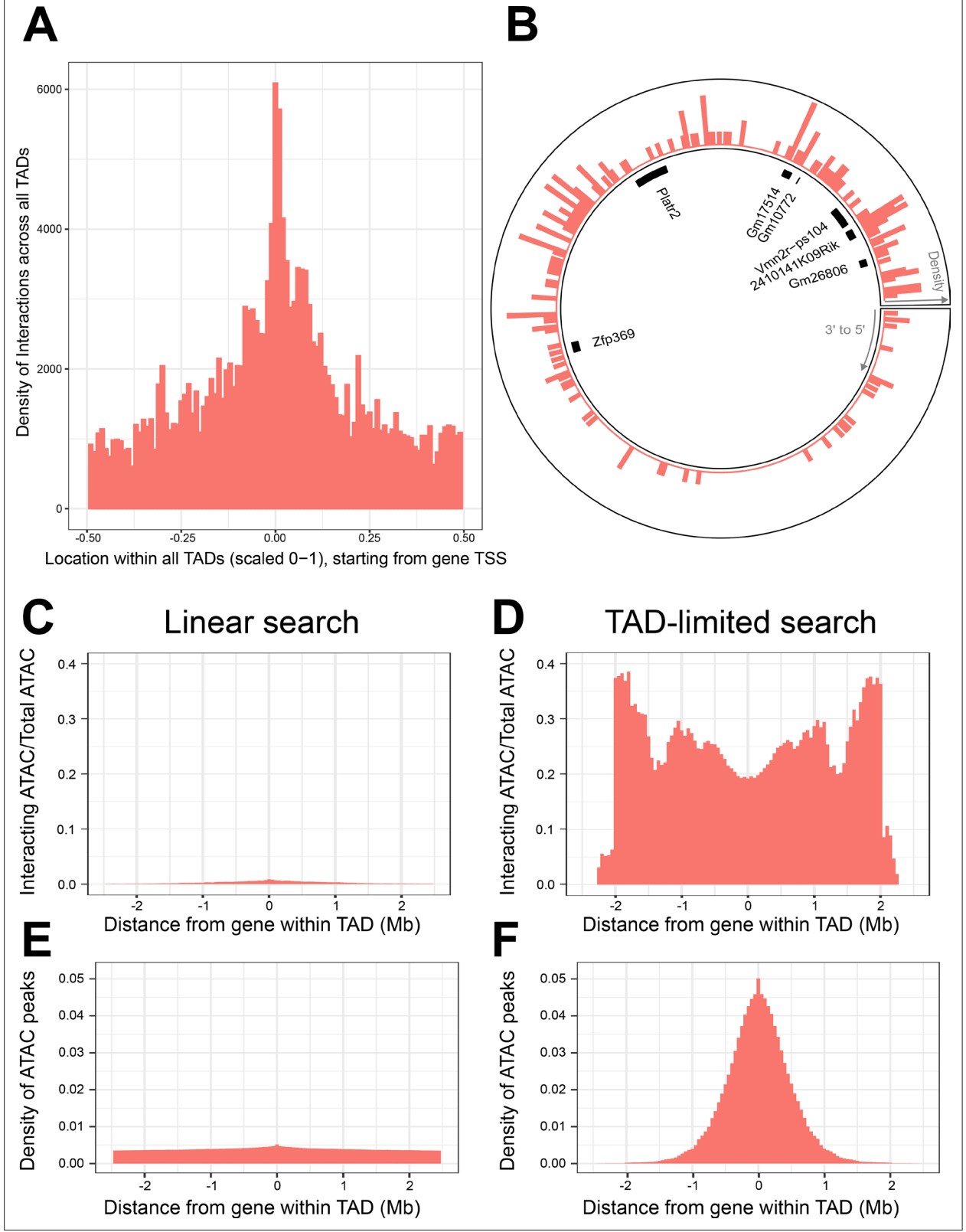

**Figure 4.** Topologically associated domains (TADs) provide context for interactions and increase interaction search efficacy. (**A**) Counts of intra-TAD ATAC-seq peaks involved in all non-additive interactive models, centered on the transcription start site (TSS) of the gene affected by the genotype–ATAC interaction. Coordinates transformed to a standard scale. (**B**) Example TAD, displaying interacting ATAC peak density and gene locations. Peak relevance generally decays relative to intra-TAD distance rather than linear chromosomal distance. (**C–F**) A comparison between linear sequence-based

*Figure 4 continued on next page*

*Figure 4 continued*

and TAD-limited search methods for interacting ATAC-seq peaks. (**C, D**) compare percentage of significantly interacting ATAC-seq peaks at each gene-relative locus. (**E, F**) compare density of ATAC-seq peaks at each locus. TAD-based search shows a higher density of interactions and places limits on search distance due to testing only TAD-internal ATAC-seq peaks.

The online version of this article includes the following figure supplement(s) for figure 4:

**Figure supplement 1.** Interacting ATAC-seq peaks do not correlate with enhancers, while topologically associated domain (TAD) boundary locations favor gene proximity.

Our results indicate that the local area for non-additive genetic–epigenetic interaction is not only constrained by TADs, but also shaped according to the overall 3D genome structure of the TAD loop. Analytical methods which reflect this are more efficient at discovering interacting regulatory elements.

## Genetic–epigenetic interactions influence gene expression more than epigenetic factors alone

The coefficients estimated by our regression model quantify the effects of the genetic and epigenetic features on the expression of each gene (*Equation 1*, Methods). We postulated that our ATAC-seq, genotypic variants, and non-additive interaction effects would display patterns that corresponded to positive and negative regulatory roles with varying strengths.

In previous analyses of SNP–SNP interactions, the importance of relative magnitude of interaction effects has been a subject of debate, as single-effect terms (main effects) are normally of greater magnitude than interaction effects (*Tyler et al., 2016*; *Leon and Heo, 2009*). However, in the context of non-additive genetic–epigenetic interactions we observed interaction effects of greater absolute value than ATAC-seq effects 61.24% of the time, greater than genotypic variant coefficients 11.42% of the time, and greater than both main effects 8.52% of the time.

This suggests that ATAC-seq peaks are generally a modifier of underlying genetics, so the majority of ATAC-seq peak effects are smaller than interaction effects. This was supported by the relatively low contribution of ATAC-seq peaks to single-effect regression models (*Figure 2D*). Their quantitative variation produces relatively subtle effects. This stands in comparison to the binary presence or absence of variants, which also directly capture regulatory sequence.

The directionality of effects on gene expression specifies the positive or negative influences from genetic variants, ATAC peaks, and their combinations. Positive and negative effects represent correlation or anti-correlation of variant presence or ATAC-seq score with gene expression. When all effects have the same sign (positive or negative), the total effect is synergistic and the pairwise combination of non-reference genotype and ATAC peak alter expression beyond the sum of the two. When the interaction effect has the opposite sign of the main effects, this indicates redundancy or interference, based on the idea that redundant factors create an 'or' logic that yields a combined result that is less than the sum-based expectation. Alternatively, mixed main effects with interactions can also signify a suppression of one outcome in favor of the other, in which the sign of the interaction term serves to move the additive expectation nearer to one of the two marginal outcomes.

We found that models indicating redundancy or interference were the most common overall, totaling 40.53–39.54% of all models (*Supplementary file 6*). Synergistic effects were found in 17.02% of all models. These two observations, suggest that a greater proportion of regulatory non-additive interactions attenuate gene expression, rather than strengthening it. While ATAC-seq peaks are often correlated with increased gene expression (*Cao et al., 2018*), we were surprised to find that an increase in ATAC-seq signal had a negative effect on gene expression in 40.76% of models (*Supplementary file 6*). Due to the high proportion of ATAC-seq peaks found within gene bodies and the association between open chromatin and gene transcription, this result warranted further investigation.

## Motif enrichment analysis reveals CTCF complex participation in genetic–epigenetic interactions

We looked next for potential binding sites or functional motifs underlying our results to provide clues as to the mechanistic underpinnings of model effects. We were especially interested in the subset of ATAC-seq peaks with negative effects on gene transcription, as these results were counter to our

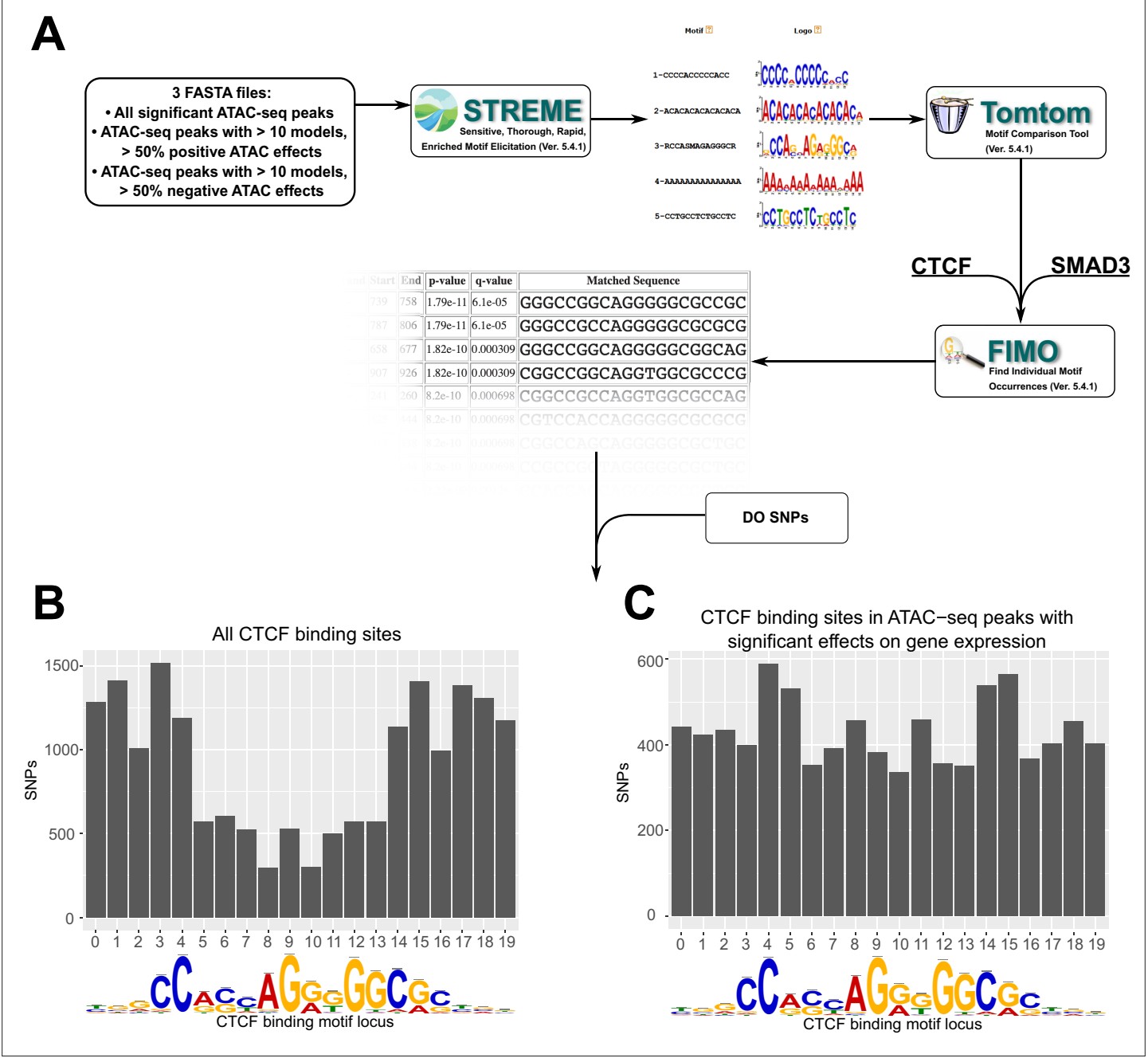

**Figure 5.** Motif analysis identifies differences in interacting CTCF-binding motifs. (**A**) A schematic of our motif analysis through MEMEsuite. FASTA files derived from interacting ATAC-seq peaks are used to identify enriched motifs, identify protein-binding sequences, and locate the sequences within the ATAC-seq peaks. (**B, C**) Binding sites found within significant motifs are less protected from genetic variation. Single-nucleotide polymorphism (SNP) counts are shown at each locus in the CTCF-binding sequence, comparing motifs within interacting ATAC-seq peaks versus all CTCF-binding sites.

The online version of this article includes the following figure supplement(s) for figure 5:

**Figure supplement 1.** Motif analysis identifies CTCF- and Smad3-binding motifs in example topologically associated domain (TAD).

**Figure supplement 2.** Analysis of relative effect magnitudes indicates multiple genetic–epigenetic interaction subtypes.

expectations. We hypothesized that these areas of open chromatin might expose binding sites of repressive regulatory factors.

To test this hypothesis, we tested for DNA motif enrichment analysis using MEMEsuite (*Figure 5A*). We selected the subset of ATAC-seq peaks involved in 10 or more significant non-additive interactions,

at least 50% of which have negative ATAC-seq effects (negative effectors). This subset was compared versus all ATAC peaks and against shuffled control sequences via STREME analysis, which finds enriched ungapped motifs in provided sequences.

Results showed the negative effector subset was enriched for 49 motifs (*Supplementary file 7*), including the CTCF-binding site (p < 2.0 × 10⁻¹⁴) and SMAD3-binding site (p < 1.5 × 10⁻⁸). These motifs were present, but less significantly enriched in positive effectors, or in all significant ATAC-seq peaks. This suggested that a portion of negative ATAC-seq effects can be functionally explained by altered behavior of CTCF-binding sites carried by specific ancestries in the DO. To complement this analysis, we also quantified the CTCF motif occupancy in negative effectors versus other ATAC-seq peaks. FIMO motif scanning showed that 53.3% of top negative ATAC-seq sequences had at least one CTCF-binding motif in them, compared to 35.3% in ATAC-seq peaks with positive effectors on gene transcription (see deposited data).

We next examined whether these peak locations contain SNPs that might alter CTCF-binding potential. Imputing from the founder genomes of the DO population, we analyzed the locations of SNPs in CTCF-binding motifs associated with regression models versus all CTCF motifs. Out of CTCF motifs within this negative subset, 96.1% of these were found to contain SNPs from the non-reference DO founder strains, versus 18.0% for positive effect-associated CTCF motifs (p < 2.2 × 10⁻¹⁶). Similar results were found for Smad3. We also found that across all motifs, the density of SNPs favored the start and end of the binding sequence (*Figure 5B*). However, in motifs associated with regression models, the density of SNPs was approximately equal across all bases (*Figure 5C*). These bases have previously been identified as having high protein–DNA-binding energies with the canonical sequence (*Cao et al., 2018*; *Zuo et al., 2017*).

Most genomic sequences matching the CTCF-binding motif are not known to be bound, according to ChIP-seq experiments (*Maurano et al., 2015*). We wanted to determine the overlap between CTCF-binding sites found in our data and known active binding sites in mESCs. Comparing to available ENCODE CTCF ChIP-seq in C57BL/6 Bruce4 and 129/Ola E14TG2a.4 mESCs (*ENCODE Project Consortium, 2012*; *Shen et al., 2012*), we found 2.04% and 1.87% of these overlapped with negative ATAC effect CTCF-binding sites that contained SNPs, versus 17.45% and 16.20% overlapping with positive effect CTCF-binding sites that contained SNPs. Expanding the scope to attempt to find flanking Smad3 regions netted consistently low results, with a 1-kb flanking window returning between 1.20% and 0.41% for Bruce4 and E14TG2a.4, respectively. These findings show that the majority of CTCF-binding sites found within our significant models are not captured in previous analyses of ESCs two different *M. musculus musculus* strains.

## Putative developmental regulator Platr2 is regulated by multiple redundant elements

To provide an example of our analytical method and probe a gene previously proposed to be important in stem cell development, we examined *Platr2* and its TAD. *Platr2*'s TAD contains a high concentration of confidently called regression models in our analysis (*Supplementary file 8*), with 1031 models of non-additive genotype–ATAC interaction reaching our significance cutoff for resident genes, of which 179 were models of *Platr2* expression. It also contains 22 haplotype markers, allowing a certain level of model variant localization. Previous studies have found a group of genes regulated in trans by expression quantitative trait loci (eQTLs) mapped to *Platr2* (*Skelly et al., 2020*). These target genes are associated with embryonic ectoderm, indicating *Platr2* may act as a regulator of stem cell state. These factors made it a target of interest for further exploration.

When the direction of effects for *Platr2* was analyzed, the distribution showed a shift toward models where ATAC and genotype effects agreed with each other, but not with the interaction term (*Supplementary file 8*). As discussed above, this potentially indicates functional redundancy between haplotype and chromatin openness at these sites. Motif enrichment analysis of interacting ATAC-seq peaks identified a sequence at 16 sites which contain Smad3-binding motifs, and another sequence at 15 sites that contain CTCF-binding motifs (*Figure 5—figure supplement 1A, B*, *Supplementary file 9*), which may suggest modulation of CTCF-binding strength. These results suggest that *Platr2* may have differential regulation patterns governed by changes in TAD formation.

## CTCF binding in inbred mESCs validates strain-specific effects

CTCF binding to DNA is associated with open chromatin around the binding site, TAD formation, and regulation of local gene expression (*Oomen et al., 2019*; *van Ruiten and Rowland, 2021*; *Li et al., 2019*; *Andrey et al., 2013*). Genetic variation within CTCF-binding sites can alter binding potential (*Cao et al., 2018*; *Zuo et al., 2017*). We hypothesized that our regression models indicated areas of strain-specific CTCF binding due to polymorphisms in the DO haplotypes. To test this, we performed CTCF ChIP-seq on mESCs derived from four of the eight DO founder strains, including representatives from the three subspecies contributing to the DO population. C57BL/6J was also included as the standard reference.

Sample ChIP-seq-binding intensities clustered by strain with principal component analysis (PCA), and were separated by subspecies as expected (*Figure 6—figure supplement 1*). Consensus peaks were identified from replicates of each strain. Across all samples, 65,541 ChIP-seq peaks passed our significance threshold (found in at least two samples in at least one strain), or 68.21% of results. 90.65% of these overlapped with the publicly available TAD dataset we had employed for our previous analyses.

Our DO data suggested to us that there are genetically determined differential CTCF-binding sites within that population. If that is the case, then we would expect to see strain-specific differences in CTCF peak occupancy. Consensus peaks were identified from replicates of each strain. Fifty-two percent of CTCF ChIP-seq peaks were shared across all four strains, with the remaining 48% found in three or fewer strains (*Figure 6A*).

We found 23,887 of our significant CTCF ChIP-seq peaks these overlapped with CTCF-binding sites that had been previously identified, or 36.45% overlap with significant results. We also found that 16,951 (25.86%) significant results overlap with the CTCF ChIP-seq data previously retrieved from ENCODE, and among those, 7880 (12.02%) that overlap with significant, interacting ATAC-seq peaks in our DO mESC data. About 80% of these (~6300) are found in all four strains.

To further explore strain-specific differences in CTCF binding, we examined reported binding intensity. Our CTCF ChIP-seq was performed as a bulk analysis, so binding intensity can show the probability of finding a protein bound at a locus within a sample. If CTCF binding is disrupted by strain-specific genetic factors, then its binding intensity will vary.

Our analysis found a range of binding intensities at individual loci, with 13% having a variance in fold enrichment greater than 10 (*Figure 6B*). This gave us confidence that there was truly differential binding of CTCF in our samples. Given that we previously saw an enrichment of CTCF-binding sites under significantly interacting ATAC-seq peaks, this lends weight to our hypothesis that we were seeing evidence of 3D chromatin structural differences in our genetically diverse mESCs.

Fifty-two percent of CTCF ChIP-seq peaks were shared across all four strains, with the remaining 48% found in three or fewer strains (*Figure 6A*). A range of binding intensities was also found at individual loci, with 13% having a variance in fold enrichment greater than 10 (*Figure 6B*).

## Non-additive interactions are predictive of CTCF-binding patterns

With our ChIP-seq data indicating the presence of strain-specific CTCF binding, we hypothesized this differential binding could be predicted from our regression models. We anticipated that inbred founder strain CTCF ChIP-seq would have more genotype effects at binding sites with open chromatin in non-additive interactions with local genotype, as opposed to binding sites without interactions.

Both interacting and non-interacting areas contain genetic variation and ATAC-seq peaks, but our models indicated interacting areas had more effects on local gene expression and genetic variability in CTCF-binding sites. More specifically, we expected negative effector ATAC-seq peaks to have the greatest predictive power, as we believed their negative effects on local gene expression arise from strain-specific differences in CTCF binding and loop formation, whereas positive effects on gene expression may come from non-localized genetic variants in other binding sites.

To test these hypotheses, we retrieved CTCF ChIP-seq locations and identified biallelic SNPs found in the four inbred DO founder strains we used for our ChIP-seq experiments. The correlation of SNP genotype to CTCF-binding site occupancy was assayed using a paired sample Student's *t*-test (*Figure 6C*).

The proportion of results below $p < 0.05$ in all CTCF ChIP-seq peaks was 16%, establishing a baseline of predictive power for SNPs genome-wide. Subsetting to those CTCF-binding sites found

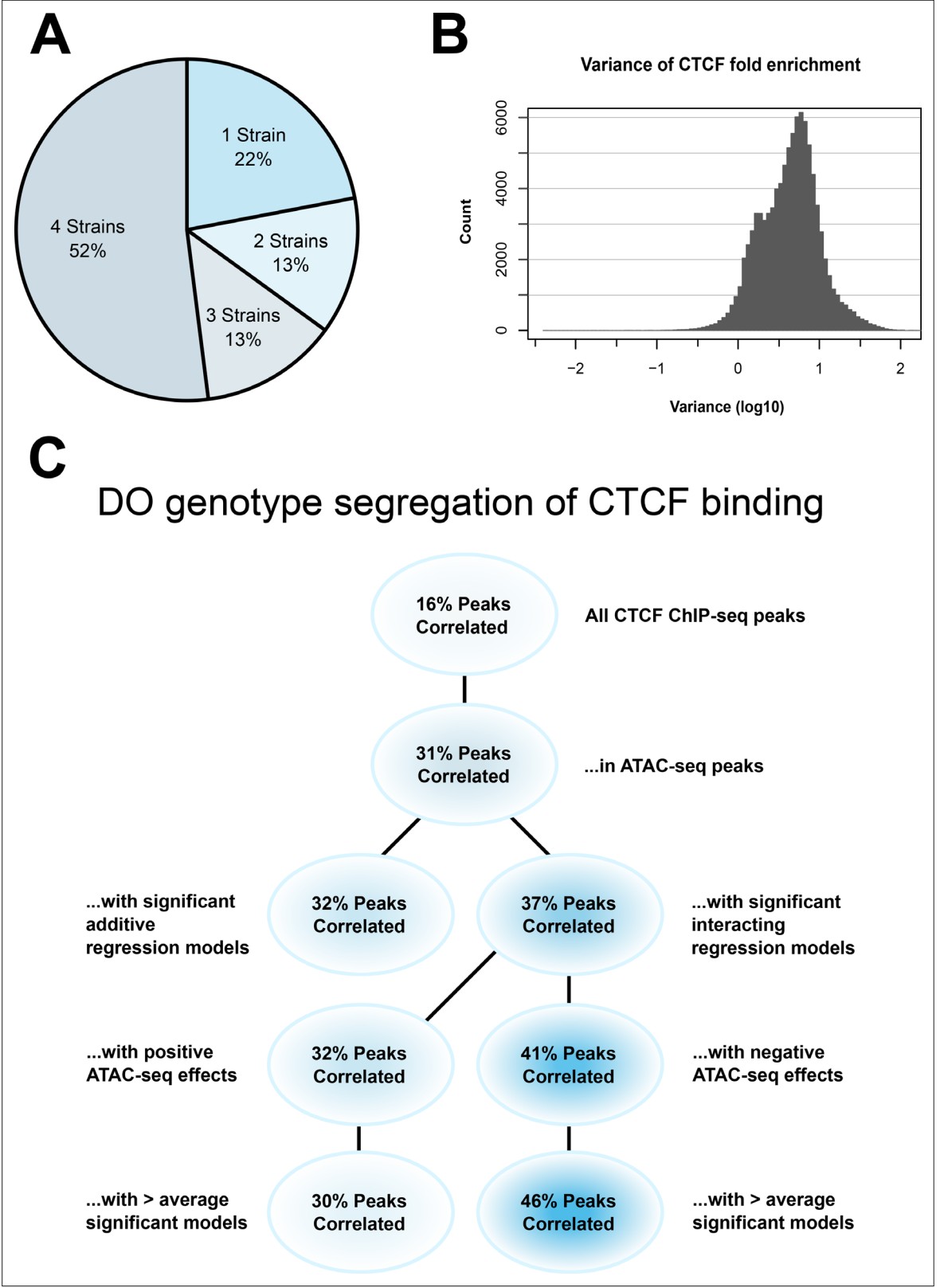

**Figure 6.** CTCF ChIP-seq analysis shows predictable strain-specific differences in binding intensity. (**A**) Percentage of ChIP-seq peaks in surveyed strains. (**B**) Variance (log10) in binding intensity fold enrichment for all ChIP-seq peaks. (**C**) Percentage of significance in association between DO genotype at CTCF peaks and CTCF-binding intensity on inbred ChIP-seq samples, in various subsets.

*Figure 6 continued on next page*

*Figure 6 continued*

The online version of this article includes the following figure supplement(s) for figure 6:

**Figure supplement 1.** Samples cluster by subspecies and strain, not by batch.

in ATAC-seq peaks increased the proportion of significant correlation to 31%. This was expected, as open chromatin in the vicinity of CTCF-binding sites is associated with binding site occupancy (*Oomen et al., 2019*; *Li et al., 2019*).

To test whether our theory that non-additive interacting regression models held greater predictive power of CTCF-binding intensity, we subset CTCF-binding sites in ATAC-seq peaks with significant effects on local gene expression. Binding sites associated with non-additive interacting models had 37% correlation. This outperformed additive models, which had 32% correlation.

ATAC-seq peaks can be associated with significant effects on multiple genes, potentially in combination with multiple genetic variants, resulting in some ATAC-seq peaks associating with multiple significant regression. Interestingly, we found that among ATAC-seq peaks over CTCF-binding sites, those associated with additive models were a subset of ATAC-seq peaks with non-additive interacting models. This means that any ATAC-seq peaks that had effects on local gene expression and were localized to CTCF-binding sites *always* had interaction effects with the local genotype, and these models are more predictive of CTCF-binding intensity.

These results matched our expectations. ATAC-seq peaks co-localized with a functional polymorphism affecting a CTCF-binding site would be more likely to affect gene expression in a non-additive fashion, as the polymorphism would only affect CTCF binding based on chromatin state at the binding site or any nearby priming factors. This is in contrast to additive models, where genetic effects and ATAC-seq effects are predicted to be independent of each other and are thus less likely to be co-localized.

Subsetting those ATAC-seq peaks that were negative effectors in interacting regression models produced a 39% correlation. This outperformed ATAC-seq peaks with a positive effect, which had 32% correlation. This was in line with our previously stated predictions and suggested that non-additive interactions can be used to evaluate and predict local 3D chromatin structure.

Finally, we wanted to determine whether the areas where we identified the most DO regression models supported our hypothesis, or if a larger amount of activity would result in a leveling effect on DO genotype's predictive power on CTCF binding. We subset the results to those regions associated with a greater than average number of regression models. Results showed that the gap between negative and positive effector regions widened in these regions to 44% versus 27% correlation. This indicates that our interacting regression model can provide detailed and biologically data in regions of dense genetic–epigenetic interaction, and further emphasizes our previous results.

## Discussion

Integrating analysis of multi-omics data in a genetically diverse population allows for greater specificity and modeling of complex regulatory interactions. Populations that segregate natural genetic variation provide dense, randomized perturbations from which gene expression variance can be modeled. This contrasts with experimental designs that rely on a limited number of engineered perturbations in isogenic cells or animals. By collecting genetic, epigenetic, and transcriptomic data from the same cellular panel, models can be inferred without confounding factors such as different experimental protocols and environments. Our approach to integrating genetic and functional genomics data from 176 DO mESC lines allowed us to systematically probe how genetic and epigenetic variation interact to jointly influence local gene expression.

We integrated ATAC-seq peaks and genotypic variants via an interactive regression model of their effects on gene expression to determine how often these factors are independent of each other, and how often they interact. From this, we were able to infer the wide-scale presence of local interactions between these regulatory mechanisms. This method is well suited for outbred populations, which are not compatible with methods used for analyzing inbred cohorts, such as differential expression or variance between samples. While similar interactions have been observed in isolated contexts and in co-localized genetic variants and ATAC-seq peaks (*Krijger and de Laat, 2016*; *Kumasaka et al.,*

2016), this marks the first time that the phenomenon has been observed genome-wide, with the genetic mapping resolution provided by outbred, heterozygous samples.

From our analysis of genetic–epigenetic interactions we have put forth several key findings regarding the interactivity, structure, and function in the regulation of gene expression. We discovered that patterns of genetic–epigenetic interaction reflect the structure of topologically associating domains. Our model inferred the presence of frequent interactions between genotypic variation and open chromatin, with a strong preference for coordinates internal to known TADs in mESCs. We demonstrated this inference in several ways, including the clustering of highly interacting areas of open chromatin within TADs, and clustering of interactions based on inter-TAD distance rather than linear DNA sequence.

Interacting ATAC-seq peaks were found to cluster within gene bodies, promoters, and annotated enhancers. These findings confirm that TADs generally define the local area for interactions. While open chromatin density is higher within TADs, we found that open chromatin density is less important for interactivity than TAD boundaries are. Furthermore, linear proximity cannot be used to identify association between genes and regulatory features. Genetic variants or areas of open chromatin that are segregated by a TAD boundary may be incapable of affecting expression of a nearby gene. Conversely, a distant genetic variant or ATAC peak may be placed near a gene within a TAD loop.

Interaction effects were further classified based on the magnitude and direction of their effects on gene expression. These models contained an unexpectedly high proportion of increased ATAC-seq peak effects associated with reduced gene expression. Upon further investigation, these were found to be enriched for CTCF-binding sites, which were further enriched for genetic variation in DO mice, particularly in the core sequence of CTCF-binding sites (*Figure 5B, C*). This indicates that these CTCF-binding sites may be differentially bound in different samples, resulting in downregulation of transcription, either by cutting off access to nearby enhancers or by abolishing TAD structure.

We further analyzed the interplay between direction of effect for genetic variants, ATAC-seq peaks, and their interactive components (as in *Figure 4—figure supplement 1A*), allowing us to make hypotheses about the functional significance of various interactions. Models where all effects are positive or negative suggest the open chromatin region and genetic variant enhance each other's effectiveness in increasing or decreasing gene expression. These synergistic effects are indicative of two regulatory factors working together to produce a greater change in gene expression, beyond what either could produce independently. Functional redundancy or interference can be inferred from models where the non-reference genotype and ATAC-seq peak have a positive effect on gene expression and the interaction effect is negative, or vice versa. Redundancy is rather common in our genetic–epigenetic interactions, while synergistic effects are relatively rare. This appears to align with previous analyses of purely genetic interactions in other mouse crosses (*Dixon et al., 2012*; *Tyler et al., 2016*). Other models are more cryptic, but contain variation in beta coefficients associated with these models (*Figure 5—figure supplement 2*), suggesting that functional subtypes that can be investigated within each group.

Our approach can be directly extended to specific gene targets. Mediation analysis in the DO by Skelly et al. revealed several mESC genes that act as mediators of downstream gene networks (*Skelly et al., 2020*; *Skelly et al., 2019*). Our analysis identified several genetic–epigenetic interactions in the area of one mediator gene, *Platr2*, and provided a targeted list of SNPs and ATAC-seq peaks that may influence the gene's expression (*Figure 4B*). ATAC-seq peaks were enriched for Smad3-binding sites, a component of the CTCF-binding complex. This provides further regulatory information and potential targets for experimental manipulation.

Through DNA motif analysis, we identified distal regulatory activity via our interaction modeling. This permits functional analysis of protein factors on gene activity that is otherwise undetected by this method, such as regulatory proteins with no local QTL. This means that while interaction models may achieve the best resolution in areas of low linkage disequilibrium, they can still be used to identify and infer regulatory action of key conserved proteins in trans.

Genetically, diverse populations in mice still contain significant linkage disequilibrium blocks which affect genetic resolution. This limited our ability to identify the causal SNPs for the purposes of this paper, but as demonstrated with our analysis of *Platr2*'s local regulatory area, the variable genetic distance across the DO genome sometimes permits us sufficient resolution to analyze whether interacting elements are co-localized or not.

Our CTCF ChIP-seq experiments identified strain-specific differences in CTCF binding, which were most predictably affected by genetic variation in regions tied to non-additive interacting regression models in DO mESCs. We also found that previously identified CTCF-binding sites in negative effector ATAC-seq peaks are more likely to contain SNPs that can predictably CTCF-binding potential.

Our results have several implications for genetic analysis of gene expression in genetically diverse populations. Experiments performed in cell cultures or isogenic models often fail to produce replicable results in other tissues or human trials, Genetic–epigenetic interactions may underpin some of these failures, particularly those that have initially shown strong genotype-dependent effects. Thus, it is important to consider generating datasets with a combination of genetic mapping, gene expression, and appropriate epigenetic data, either on a local or genome-wide scale. At present, publicly available data that match these criteria are not available. As research expands toward greater coverage in wild populations and humans, accessibility is likely to increase. Furthermore, ATAC-seq could be substituted for other experimental datasets, such as H3K4me3 ChIP-seq.

In the future, our study would benefit from the generation of more precise mapping of active CTCF-binding sites, across more genetic backgrounds. Our CTCF ChIP-seq experiments expanded on previous findings that mouse strains have different strengths in CTCF binding (*Fasolino et al., 2020*). Our results indicated that almost half of CTCF binding is strain specific in mESCs, with imputed links to genetic variants in the DO mouse population. We also found a link between CTCF-binding site occupancy and our imputed genetic–epigenetic interactions, which could indicate 3D structural variants that have effects on local gene expression.

To confirm this, more experiments are needed. ChIP-seq can identify where CTCF is binding, but not whether it has formed a TAD-binding complex. Chromatin conformation experiments like Hi-C and ChIA-PET in the DO founder strains would permit us to directly examine genome structure. This could determine whether there are interactions that were obscured in some samples by the blocking effect of a TAD boundary, and whether there are some TADs that are more tolerant of boundary shifts than others.

Small changes in TAD boundary location are previously indicated to be involved in developmental and differentiation processes (*Rodríguez-Carballo et al., 2020*; *Andrey et al., 2013*; *Takayama et al., 2021*), and other studies have shown that chromatin rearrangement that alters TAD boundaries is linked to cancer development (*Dixon et al., 2012*; *Tyler et al., 2016*; *Menghi et al., 2018*). However, many publicly available datasets of CTCF-binding activity have limited resolution and/or coverage. With CTCF-binding sites scattered across the genome, there could be further subtle shifts in local regulatory areas that have previously gone unnoticed, particularly in developmental contexts or across different tissues.

Overall, this study demonstrates that genetic–epigenetic interaction analysis can reveal 3D genome structure through the positioning of interacting genome features. We see in this how genetics and genome structure can inform each other. These findings imply that including TAD boundaries and TAD loops in analyzing genomic features affecting gene expression, such as chromatin states and genetic variants, can maximize and contextualize results.

## Methods
### Sample production and initial processing
DO mESC production was performed by Predictive Biology. Cultures were grown with G3SK inhibitor (1i medium). Bulk RNA and ATAC sequencing were performed and normalized as previously described (*Skelly et al., 2020*). RNA-seq resulted in 6M-55M $2 \times 75$ bp paired-end (PE) reads per sample. ATAC-seq libraries were amplified for nine cycles and purified with 1.7X AMPure beads. Expression data were formatted and analyzed as log2-transformed transcripts per million (TPM). ATAC-seq data were formatted as trimmed mean of $M$ values (TMM) (*Robinson and Oshlack, 2010*). Genotyping was performed by Giga Mouse Universal Genotyping Array (GigaMUGA), an Illumina Infinium array of 143,000 SNP markers with a special focus on DO founder strains (*Morgan et al., 2015*). Aneuploidies were removed with the argyle R package, by identifying chromosome-level gene expression differences (*Morgan, 2015*). QTL2 haplotype reconstruction, normalization, and pseudoprobe processing were carried out as previously described (*Skelly et al., 2020*). Samples with X0 genotypes were removed, and the union of all samples with the required data types resulted in 176 samples.

Sample genotypes were estimated in reflection of linkage disequilibrium and haplotype prediction confidence, employing a previously calculated grid of GigaMUGA pseudoprobes, which were evenly spaced by genetic distance (*Bubier et al., 2020*, *Chick et al., 2016*). This produced a subset of 68,413 imputed haplotype calls centered at specific loci which were most likely to match the actual haplotype in each region (Supplementary Information).

## Regression modeling

RNA-seq, ATAC-seq, and haplotype data were processed into SQLite databases and fit to a regression model using the stats::step() function in R. We modeled the expression of each gene by linear regression with a genotype-by-ATAC non-additive interaction term:

$$y_i = \beta_0 + \beta_1 x_1 + \beta_2 x_2 + \beta_3 x_1 x_2 + \varepsilon_i \tag{1}$$

where $y_i$ = RNA abundance (log$_2$TPM), $x_1$ = genotype, and $x_2$ = ATAC intensity (TMM). Genotypes were taken from the haplotype estimations, and coded as 0 for reference, 1 for heterozygote, and 2 for homozygote non-reference. Multi-allelic variants were not present in the pseudoprobes and were therefore not accounted for in the method. Models that retained the $\beta_3 x_1 x_2$ term by passing the default Akaike information criterion cutoff were deemed to be interacting. These were used as the input for all further analyses.

This regression model was compared to a null model in which a genetic variant and open chromatin element affect gene expression, but do so independently of each other ($\beta_3 = 0$ in *Equation 1*). F statistics were calculated with the R function pf(). Given the possibility of overfitting as the number of model terms increases and the high number of models tested (see Results), a Bonferroni adjusted p value cutoff of p < 1 × 10$^{-7}$ was selected (hereafter referred to as 'significant models'). Bonferroni corrections were calculated via stats::p.a just().

## Topologically association domain, motif, and DNA-binding analyses

mESC TAD data were derived from previously published work (*Dixon et al., 2012*). in a liftover from mm9 to mm10 via UCSC genome browser (*Kuhn et al., 2013*). Chromosome information for mm10 retrieved from the Integrative Genomics Viewer (*Robinson et al., 2011*).

DNA motif and binding site discovery were performed using the MEMEsuite tools MEME, STREME, and Tomtom, all set to default parameters (see Supplemental Data) (*Bailey et al., 2009*; *Bailey and Elkan, 1994*; *Bailey, 2020*; *Gupta et al., 2007*). SNP data were imputed from DO founder genomes referenced from release v3 (REL-1303-SNPs_Indels-GRCm238) of the Mouse Genome Project through the Sanger Institute (*Grant et al., 2011*; *Keane et al., 2011*).

We downloaded the CTCF ChIP-seq in C57BL/6 Bruce4 and 129/Ola E14TG2a.4 mESCs from the ENCODE portal (*Sloan et al., 2016*) with the following identifiers: ENCSR000CCB and ENCSR362VNF (*ENCODE Project Consortium, 2012*). These were also used to estimate the locations of Smad3 activity. CTCF- and SMAD3-binding motifs were retrieved from HOCOMOCO (*Kulakovskiy et al., 2018*) with the identifiers CTCF_MOUSE.H11MO.0.A and SMAD3_MOUSE.H11MO.0.B. These were overlapped with our ATAC-seq dataset via FIMO (*Grant et al., 2011*).

## Cell lines

All cell line identities were authenticated using The Jackson Laboratory's Genetic Quality Control Program. This Program includes a 27-SNP panel which was used for strain confirmation. Additionally, all cell lines tested negative for mycoplasma, using a qPCR-based assay, and bacterial and fungal contaminants.

## CTCF ChIP-seq

We performed CTCF ChIP-seq following previously established protocols (*Oomen et al., 2019*). Three cell lines were run for four isogenic mESC strains, C57BL/6J, CAST/EiJ, PWK/Ph, and WSB/EiJ. Cultures were grown in 1i medium. One sequencing run failed, resulting in the loss of one PWK and one WSB sample.

FastQC, Bowtie alignment, and g2gtools liftover from imputed strain genomes to BL/6J were performed using a modification of a now publicly available ATAC-seq pipeline, accessed in development

December 15, 2020 (*Lloyd, 2024*). Mitochondrial read filtering steps were removed. Results were filtered to those ChIP-seq-binding locations that appeared in two or more isogenic strains.

All study data and scripts are available at Synapse (https://doi.org/10.7303/syn26534443).

## Acknowledgements

We thank G Churchill for helpful comments. This study was funded by the National Institute of General Medical Sciences of the National Institutes of Health under award number R01GM115518. CLB was supported by the National Institute of General Medical Sciences (NIGMS) grant R35GM133724. LGR was funded in part by National Institutes of Health award P40OD011102. The content is solely the responsibility of the authors and does not necessarily represent the official views of the National Institutes of Health.

## Additional information

### Funding

| Funder | Grant reference number | Author |
| --- | --- | --- |
| National Institute of General Medical Sciences | R01GM115518 | Gregory W Carter |
| National Institute of General Medical Sciences | R35GM133724 | Christopher L Baker |
| National Institutes of Health | P40OD011102 | Laura G Reinholdt |

The funders had no role in study design, data collection, and interpretation, or the decision to submit the work for publication.

### Author contributions

Lauren Kuffler, Conceptualization, Data curation, Software, Formal analysis, Validation, Visualization, Methodology, Writing - original draft, Project administration, Writing – review and editing; Daniel A Skelly, Resources, Data curation, Formal analysis; Anne Czechanski, Resources, Investigation; Haley J Fortin, Validation, Investigation, Methodology; Steven C Munger, Resources; Christopher L Baker, Laura G Reinholdt, Resources, Supervision, Funding acquisition, Validation, Methodology; Gregory W Carter, Conceptualization, Formal analysis, Supervision, Funding acquisition, Methodology, Project administration, Writing – review and editing

### Author ORCIDs

Lauren Kuffler ⓘ http://orcid.org/0000-0002-4991-2292
Daniel A Skelly ⓘ http://orcid.org/0000-0002-2329-2216
Haley J Fortin ⓘ http://orcid.org/0000-0003-4541-0874
Christopher L Baker ⓘ http://orcid.org/0000-0001-6677-0356
Laura G Reinholdt ⓘ http://orcid.org/0000-0003-4054-4048
Gregory W Carter ⓘ https://orcid.org/0000-0002-2834-8186

Reviewer #1 (Public Review): https://doi.org/10.7554/eLife.88222.3.sa1
Reviewer #2 (Public Review): https://doi.org/10.7554/eLife.88222.3.sa2
Author response https://doi.org/10.7554/eLife.88222.3.sa3

## Additional files

### Supplementary files

• Supplementary file 1. Counts and percentages within a database of randomly generated regression models.

• Supplementary file 2. Counts and percentages within a database of all possible regression models

where all single-nucleotide polymorphisms (SNPs) and ATAC peaks are within ±1 TAD of the gene they interact with.

• Supplementary file 3. Counts and percentages of genotypic variants and ATAC-seq peaks within ±2 Mb of the gene they are imputed to affect. In additive and interacting models, we include the percent of models in which the genotypic variant and ATAC peak are closer to each other than to the gene they affect.

• Supplementary file 4. Table providing a breakdown of interacting ATAC-seq peak locations relative to gene features.

• Supplementary file 5. Chromosome comparison of percentages of interacting ATAC-seq peaks in transcripts versus in enhancers.

• Supplementary file 6. Model percentages calculated by distribution of effect signs for all significant interacting models.

• Supplementary file 7. STREME output of motifs detected within negative effector ATAC-seq peaks.

• Supplementary file 8. Model percentages calculated by distribution of effect signs for Platr2's interacting models.

• Supplementary file 9. TOMTOM output of motifs aligned to a sequence identified as enriched by MEME within interacting significantly enriched ATAC-seq peaks.

• MDAR checklist

## Data availability

All new study data and scripts are available at Synapse (https://doi.org/10.7303/syn26534443).

The following dataset was generated:

| Author(s) | Year | Dataset title | Dataset URL | Database and Identifier |
|---|---|---|---|---|
| Kuffler L, Fortin HJ, Skelly DA, Czechanski A, Munger SC, Reinholdt LG, Baker CL, Carter GW | 2023 | CTCF ChIP-seq from mouse embryonic stem cells | https://doi.org/10.7303/syn26534443 | Synapse, 10.7303/syn26534443 |

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
