## [Editor Report · eLife assessment]

This **important** manuscript reports interactions between genetic variation, DNA accessibility, and chromatin structure in gene expression at a genome wide scale. The authors found that most of these interactions occur within topologically associating domains (TADs) and 3D genome structure data can be efficiently used to guide the discovery of significant genetic and epigenetic influences on gene expression. Overall, this **convincing** study highlights the importance of 3D chromatin structure in controlling how gene expression is regulated by genetic and epigenetic processes.

---

## [Referee Report · Reviewer #1 (Public Review)]

This is an important manuscript that links gene expression to genetic variants and regions of open chromatin. The mechanisms of genetic gene regulation are essential to understanding how standing genetic variation translates to function and phenotype. This data set has the ability to add substantial insight into the field. In particular, the authors show how the relationships between variants, chromatin, and genes are spatially constrained by topologically associated domains.

---

## [Referee Report · Reviewer #2 (Public Review)]

The experiments described in the manuscript are well designed and executed. Most of the data presented are of high quality, convincing, and in general support the conclusions made in the manuscript. This manuscript should be of great interest to the field of mammalian gene regulation and the approaches used here can have broader applications in studying genetic and epigenetic regulations of gene expression. The key finding reported here, the importance of 3D chromatin structure in controlling gene expression, although not unexpected, offers a better understanding of the physiological roles of TADs.

Comments on revised version:

I think the authors have substantially addressed reviewers' concerns. I have no further comments to add.

---

## [Author Response]

The following is the authors’ response to the original reviews.

We want to thank the reviewers for their thoughtful analysis and questions.

A brief overview of the changes to the manuscript is provided here, with individual responses to the reviewer comments following.

The methods section has been expanded to better explain the techniques used in our analyses. CTCF binding data section has likewise been expanded, to include more detail on the dataset and our analysis of its contents. All other requested clarifications have been added to areas of the results.

Beyond specific requests from the reviewers, we made the following changes.

We felt that a particular terminology choice on our part resulted in some confusion: the use of “SNPs” to refer to genetic variants within our Diversity Outbred samples. While we used SNPs that lay closest to the center of our haplotype predictions as our representative loci for each linkage disequilibrium block, this was done for computational purposes only. We did not focus most of our analyses on the haplotypes themselves, because of the uncertainty of which variants within an LD block actually participated in the genetic-epigenetic interactions we imputed.

Thus, we edited the text to remove mention of “SNPs” unless our analysis did directly and deliberately profile SNPs themselves. In all other cases, we now refer to “haplotypes”, “genetic variants”, or “variants”. This should help increase clarity in the manuscript as a whole.

A small error was discovered within the labelling and processing of regression model outputs in chromosome 14. A consistency check was run on all chromosomes, finding that only Chr 14 was affected. Chr 14 was rerun in its entirety to verify its results, with the previous results now archived within our databases uploaded on Synapse (see Methods for a link). All relevant calculations and figures were regenerated, resulting in an average shift of 1% or less across the manuscript. All analyses remain highly statistically significant.

Responses to comments from Reviewer #1

Methods

Sequencing depth was retrieved from the original publication on the primary multiomics dataset. (Line 105-106)A line was added regarding initial mouse genome alignment for the original publication: we explain the GigaMUGA genotyping array, used for the DO mESC samples. For our ChIP-seq data, we reword to specify: we used liftovers from imputed strain-specific genomes to B6 mm10. (Lines 108-110; 116-120; 168-170)

Aneuploidy removal is expanded upon in a similar fashion: the original QC identified chromosome-level gene expression differences to remove aneuploid samples. (Line 111)Mention of the pre-publication use of an alternative null model has been removed, given its lack of relevance to the rest of the text. While it was interesting to compare to the standard null model, it amounts to a side note that distracts from the focus of the paper. (Line 137-139).Descriptive subheadings have been added.

Results

Line 179 (now Line 191) now points to Methods.Line 189-200 (now Line 188-204): language altered to better explain our intent: We wished to perform an intrachromosomal scan across the whole genome for non-additive genetic-epigenetic interactions. However, there were computational limits to how many possible combinations of gene, haplotype, and ATAC-seq peak we could feasibly test. We thus generated a random subset of possible combinations. This was also performed to identify target regions for focused analyses.Line 195 (now line 206, expanded on in Line 210): Clarification added on the significance of our result: if non-additive genetic-epigenetic interactions were not a significant explanatory factor for gene expression, we would expect to see no enrichment of low p-value results. Instead, we see 0.07% of our models coming in at adj. p < 1x10-7.Line 199 (now Line 216): The requested calculations were run, and are now included in table S3. We found that within 4 Mb of a given gene, less than 10% of variants and ATAC peaks within clustered closer to each other than they did to the gene they affected.

Please note that this figure has a level of uncertainty due to linkage disequilibrium. Thus, rather than precisely answering the question “[are there haplotype-ATAC pairs] that are in the same locality but further away from the gene?”, we asked "is the ATAC peak closer than the gene to the point where we have the highest confidence of correctly calling the interacting genotype?". The relevant code has been deposited in our Synapse repository (see Methods for link).

Line 205 (now restructured in Line 221-228): The text has been edited to specify our intent. We are referring to a set of TAD-focused regression models we generated (see Methods) that comprehensively included all possible interactions between genes, and all haplotypes and ATAC peaks within +/- 1 TAD of the gene.(Line 227): We specified that the previously-published TAD boundary dataset we used was retrieved from the Bing Ren lab’s Hi-C projects, which imputed locations of TAD boundaries in B6 mESCs.We have relabeled Figure 1 and tweaked the surrounding text to clear up some confusing aspects. The Euler plots in Figure 1D-E reflect the fact that each ATAC-seq peak and haplotype can be in multiple relationships with local genes and regulatory factors. Some of these relationships will be simple correlation between their presence and gene expression, while others may co-regulate alongside independent regulatory factors, or engage in non-additive regulatory interactions.

Because these non-additive regulatory interactions have not been comprehensively studied, we wished to determine whether there were any regulatory factors within our data that would not be detected as significant via more conventional methods, such as correlation analysis, mediation analysis, or regression analysis without an interaction term. Our Euler plots show that there are large subsets of both ATAC-seq peaks and haplotypes that are exclusively found in non-additive interactions. Thus, our justification for focusing on non-additive interactions for the rest of the paper.

Line 256 (now Line 252-255): We further clarified the above in this section: correlation and mediation analyses were previously completed by the team which initially analyzed the DO mESC dataset (Skelly et al. 2020, Cell Stem Cell). They performed a correlation analysis between open chromatin and gene expression (Skelly et al. Fig. 2A), and identified expression quantitative trait loci (eQTL) (Skelly et al. Fig. 2E). We felt that more direct comparisons to the Skelly et al. data would distract readers from our focus on genetic-epigenetic interactions. Thus, we limited our discussion of non-interacting regulatory relationships to Figures 1-2, and a brief mention in Figure 5.Line 290 (now Line 337): We pulled promoter locations from the FANTOM5 database of mouse promoters, and included analysis in both the text and Figure S4A-B.(Line 475-476): we clarified “DO founder SNPs” to “SNPs from the non-reference DO founder strains”.Line 472 (restructured in Lines 531-564): We have expanded on this section, including answers to the reviewer’s questions regarding ChIP-seq peak counts, overlap with the TAD map we used for our other analyses, and expanded upon strain-specific CTCF binding we identified in our ChIP-seq analysis.

Responses to comments from Reviewer #2:

(1) Typo corrected.

(2) Lines 194-195 (now line 206, expanded on in Line 210): We have expanded upon the intent and expectations of our analysis. In summary: if non-additive genetic-epigenetic interactions were not a significant explanatory factor for gene expression, we would expect to see no enrichment of low p-value results. Thus, we would expect 0.0000001% of results to reach adj. p < 1x10-7. Instead, we see 0.07% of our models coming in at adj. p < 1x10-7, four orders of magnitude greater than expected.

(3) Lines 226-230 (Expanded on in Lines 252-276): We have relabeled Figure 1 and tweaked the surrounding text to clear up some confusing aspects. The percentages in the text are derived from the data summarized in the Euler plots in Figure 1D-E. These plots reflect the fact that each ATAC-seq peak and haplotype can be in multiple relationships with local genes and regulatory factors. Some of these relationships will be simple correlation between their presence and gene expression, while others may co-regulate alongside independent regulatory factors, or engage in non-additive regulatory interactions.

(4) Line 261-263 (now lines 299-300): A companion to Figure 2B has been added (Fig. S3), which provides interaction counts for each ATAC-seq peak that contributed to Figure 2B. A horizontal line is included to highlight the locations of the highly-interacting ATAC peaks.

(5) Analysis regarding Figure 3B had been removed from its original context. It has now been restored to the manuscript (Line 368-371).